# SARS-CoV-2 genotyping and sequencing following a simple and economical RNA extraction and storage protocol

Sarah Hernandez[1], Phuong-Vi Nguyen[1], Taz Azmain[2], Anne Piantadosi[1,2], Jesse J. Waggoner[1,3]*

1 Emory University Department of Medicine, Division of Infectious Diseases, Atlanta, Georgia, United States of America, 2 Emory University Department of Medicine, Department of Pathology and Laboratory Medicine, Atlanta, Georgia, United States of America, 3 Rollins School of Public Health, Department of Global Health, Atlanta, Georgia, United States of America

* jjwaggo@emory.edu

## Abstract

Since the beginning of the SARS-CoV-2 pandemic, supply chain shortages have caused major disruptions in sourcing the materials needed for laboratory-based molecular assays. With increasing demand for molecular testing, these disruptions have limited testing capacity and hindered efforts to mitigate spread of the virus and new variants. Here we evaluate an economical and reliable protocol for the extraction and short-term ambient temperature storage of SARS-CoV-2 RNA. Additional objectives of the study were to evaluate RNA from this protocol for 1) detection of single nucleotide polymorphisms (SNPs) in the *spike* gene and 2) whole genome sequencing of SARS-CoV-2. The RNAES protocol was evaluated with residual nasopharyngeal (NP) samples collected from Emory Healthcare and Emory Student Health services. All RNAES extractions were performed in duplicate and once with a commercial extraction robot for comparison. Following extraction, eluates were immediately tested by rRT-PCR. SARS-CoV-2 RNA was successfully detected in 56/60 (93.3%) RNAES replicates, and Ct values corresponded with comparator results. Upon testing in spike SNP assays, three genotypes were identified, and all variant calls were consistent with those previously obtained after commercial extraction. Additionally, the SARS-RNAES protocol yield eluate pure enough for downstream whole genome sequencing, and results were consistent with SARS-CoV-2 whole genome sequencing of eluates matched for Ct value. With reproducible results across a range of virus concentrations, the SARS-RNAES protocol could help increase SARS-CoV-2 diagnostic testing and monitoring for emerging variants in resource-constrained communities.

## Introduction

The COVID-19 pandemic has resulted in unprecedented challenges to global outbreak responses in healthcare systems around the world. The ongoing effort to monitor the spread of severe acute respiratory syndrome coronavirus-2 (SARS-CoV-2) and emergence of new

**Funding:** This research was supported by an award from the Doris Duke Charitable Foundation (Clinical Scientist Development Award 2019089, JJW; https://www.ddcf.org/). The funders had no role in study design, data collection and analysis, decision to publish, or preparation of the manuscript.

**Competing interests:** The authors have declared that no competing interests exist.

variants in the community relies on consistent and accurate diagnostics with paired genomic surveillance [1–3]. Despite a large increase in the development of diagnostic platforms, the most reliable molecular techniques still require highly purified nucleic acids [4]. Sample-to-answer devices address this with expensive, onboard RNA extraction, but this does not yield material for downstream variant characterization or additional testing. In developing markets or those overwhelmed with demand, the proprietary materials needed for such techniques are difficult to source and maintain [5, 6].

Our group recently developed and optimized an economical and reliable protocol for the extraction and storage of RNA from blood-borne RNA viruses (termed the RNAES protocol) [7]. The RNAES protocol capitalizes on the charge-based chemistry of RNA-silica interactions to yield eluate compatible with diagnostic real-time RT-PCR (rRT-PCR) testing. Using residual clinical samples collected from Autumn 2020 through Spring 2022, this protocol was evaluated for extraction of SARS-CoV-2, and the quality of eluted RNA was further evaluated using genotyping rRT-PCRs that detect single nucleotide polymorphisms (SNPs) in the spike gene and a common whole genome sequencing protocol. Genotyping and sequencing are essential for monitoring the evolution of pathogenic viruses yet require larger amplicons than standard detection assays, and such methods were not evaluated during the initial development of the RNAES protocol [7].

## Materials and methods

### RNAES protocol

RNA was extracted from residual clinical material using the RNAES protocol, as previously described [7]. RNAES extraction packets were assembled with a 5.56-mm diameter membrane disk sandwiched between a square blotter pad base (25 x 25 x 2.5 mm; VWR International, Radnor, PA) and a Parafilm cover (Research Products International, Mt. Prospect, IL) with a 3.96-mm diameter aperture centered over the membrane. Briefly, the protocol consisted of incubating 25µL of respiratory swab sample in 25µL of lysis mixture (150 mM sucrose, Boston BioProducts, Ashland, MA; 5µg proteinase K, New England Biolabs, Ipswich, MA; 2.5 µg carrier RNA, Qiagen, Germantown, MD; 100 mM KCl; and 50mM Tris-HCl, pH 7.0, Millipore-Sigma) for 10 minutes. Following incubation, 100µL of arginine binding buffer (100 mM L-Arginine; 400 mM KCl, both from MilliporeSigma)/150µL ethanol mixture was combined with the lysate and run dropwise through 5.56mm circular filter membranes. Membranes were washed one time with 1M glycine-HCL buffer (pH 2.7±0.1, 10X Concentrate Solution); RNA was then eluted into 50µL Tris-EDTA buffer and subsequently tested by rRT-PCR. Whatman 3, Fusion 5, and glass microfiber (GF/D) membranes (all from MilliporeSigma, Burlington, WA) were evaluated for optimal performance with SARS-CoV-2.

### Clinical samples and rRT-PCR

Residual nasopharyngeal (NP) samples collected from Emory Healthcare system and Emory Student Health services from Autumn, 2020 through Spring, 2022 were utilized for this study. Upon collection, samples were placed into saline or viral transport medium, deidentified, aliquoted, and stored at -80˚C until nucleic acid extraction. All samples had previously tested positive for SARS-CoV-2 RNA using one or more rRT-PCRs with targets in the nucleocapsid and envelope genes [2, 8, 9]. Use of anonymized residual NP samples for research performed in this study was reviewed and approved by the Emory University Institutional Review Board, and the need for consent to use these specimens was waived.

During a single freeze-thaw cycle, all samples were re-extracted in duplicate with the RNAES packets and once using the MagMaxViral RNA Isolation Kit (Applied Biosystems) in

a KingFisher Apex (ThermoFisher Scientific) commercial robotic extraction system. RNA was extracted from 25μL of sample and eluted into 50μL of buffer for both methods. For the Mag-Max extraction protocol, samples were brought up to a total initial volume of 150μL with PBS. Following extraction, eluates were immediately tested in the CDC Flu SC2 assay [10].

### Spike SNP testing

Eluates that tested positive for SARS-CoV-2 RNA in the Flu SC2 assay with sufficient volume remaining were tested in two Spike SNP rRT-PCRs, which detect mutations in *spike* associated with variants of concern and were performed as previously described [2, 8]. Samples were run in two multiplex assays that contained probes for the following mutations: 1) K417 (positive with ancestral sequence), 452R, 484K, 501Y and 2) 452Q, 478K, and 490S. Flu SC2 and Spike SNP rRT-PCRs were performed on a Rotor-Gene Q instrument (Qiagen, Germantown, MD) using 5μL of eluate and 20μL of the Luna Probe one-step RT qPCR kit (NEB), for a total of 25μL per reaction.

### Sequencing & analysis

Extracted RNA samples were treated with ArcticZymes HL-dsDNase enzyme followed by random priming and first strand cDNA synthesis using SuperScript IV (Invitrogen). Amplicon-based libraries were constructed from cDNA using xGen SARS-CoV-2 Amplicon panel (IDT) following the manufacturer's protocol. Briefly, multiplex PCR was performed on $1^{st}$ strand cDNA using SARS-CoV-2 specific primers with 18–25 cycles of amplification with a subsequent 1.0X Ampure XP bead cleanup (Beckman Coulter). Unique Dual Index primer pairs were added to 5' and 3' ends of amplicons to create ~300 bp libraries by means of Indexing PCR with 5–9 cycles of amplification followed by 0.65X Ampure XP bead cleanup. The libraries were quantified using KAPA universal complete kit (Roche), pooled to 4 nM and sequenced on Illumina Miseq with paired-end 150-bp reads. The whole genome consensus sequence was assembled using viralrecon analysis pipeline v2.4.4 [11]. Water was used as negative control.

### Stability

RNA stability at ambient temperature on dried RNAES packet membranes was the assessed in 5 samples at 0, 1, 3, and 7 days post extraction (n = 8 packets per sample; two for each time point). To establish a baseline, samples were completely extracted on day 0 and immediately tested by rRT-PCR for detection of SARS-CoV-2 RNA using an assay for the N2 target, performed as previously described [9]. This assay was selected for use in the stability analysis because our group has evaluated this as a quantitative test. For each of the remaining time points (days 1, 3, and 7), dried membranes were stored in 1.5mL tubes and placed in zipper-locked plastic bags with desiccant packets. On the day of testing, RNA was eluted from dried membranes with 50μL TE buffer and eluates were immediately tested for comparison with day 0 results. A four-point standard curve with synthesized, quantified ssDNA containing the N2 target (Integrated DNA Technologies, Coralville, IA) was included on each run to calculate SARS-CoV-2 RNA concentration at each time point.

### Statistical analysis

Calculation of means and standard deviations were done in Excel software (IBM). ANOVA and two-sided t-tests were performed in GraphPad Prism, version 9.3.1 (GraphPad Software).

## Results

### Membrane optimization

Extraction of SARS-CoV-2 RNA using the RNAES protocol was evaluated using packets constructed with Fusion 5, Whatman 3 and GF/D membranes. All samples were extracted in duplicate with the RNAES packets and immediately tested by rRT-PCR. Fusion 5 was the only membrane that resulted in successful RNA recovery for 6/6 replicates tested, yielding an average N2 Ct value of 29.45 (standard deviation (SD) 1.91). None of the samples extracted with RNAES packets prepared with Whatman 3 or GF/D membranes resulted in detectable cycle threshold (Ct) values. Based on these data, Fusion 5 membranes were chosen for the final SARS-specific RNAES protocol.

### Clinical evaluation

Thirty archived, residual samples were selected that had previously tested positive for SARS-CoV-2 by rRT-PCR. On each day of testing, samples were thawed and extracted in duplicate with the RNAES protocol (n = 60) and once for comparison with a commercial Apex extraction robot, then subsequently tested by rRT-PCR in the Flu SC2 assay. Following extraction with RNAES packets, SARS-CoV-2 RNA was successfully detected in 56/60 replicates (93.3%) and Ct values corresponded with comparator results from the commercial extraction robot (Fig 1, S1 Table in S1 File). Extraction was successful in 55/56 replicates (98.2%) with Ct values

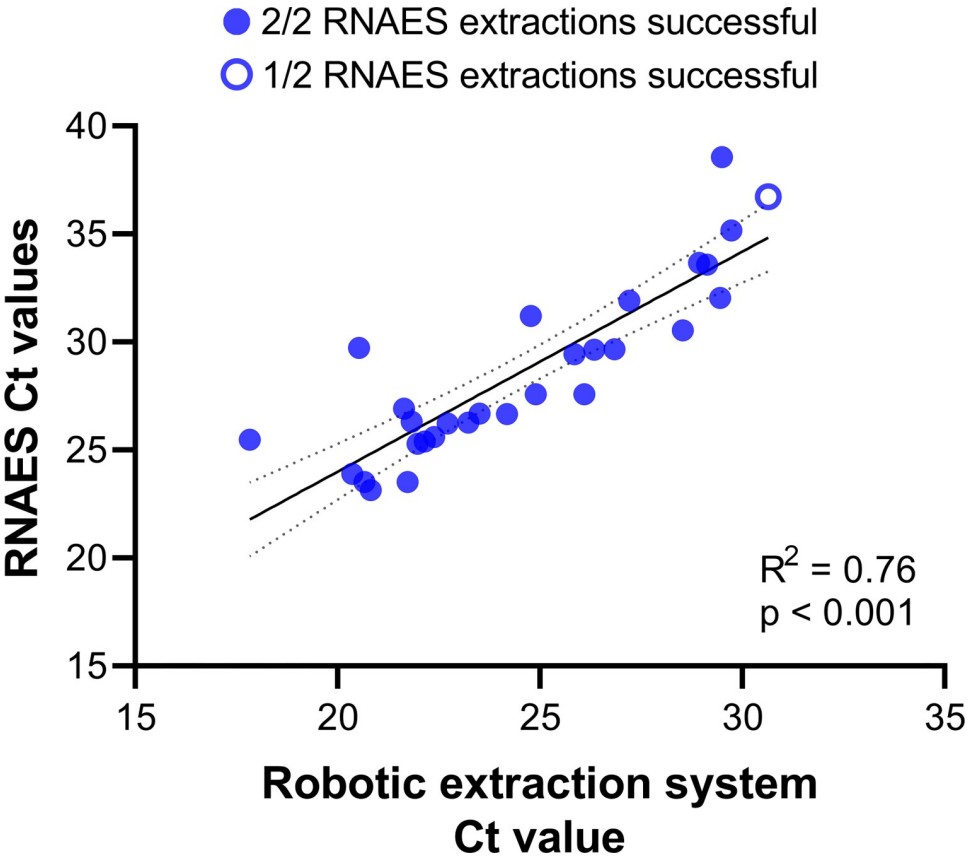

**Fig 1. SARS-CoV-2 RNA concentration in eluates from the RNAES protocol compared to the KingFisher Apex robotic extraction system.** Average Ct value for replicate RNAES extractions is displayed. Solid line displays the result of linear regression; dotted lines show the 95% confidence interval of the best-fit line.

**Table 1. Genotypes detected in the Spike SNP assay following RNAES extraction.**

| Category | n (%) | Ct, average (SD) | Lineage |
|---|---|---|---|
| Tested in Spike SNP | 53 (100) | 28.6 (4.1) | |
| Spike SNP genotype detected | 52 (98.1) | - | |
| K417 only | 19 (35.8) | 29.69 (3.31) | Ancestral |
| K417variant/478K/501Y | 31 (58.5) | 28.30 (4.64) | Omicron |
| K417/452R/478K | 2 (3.8) | 29.73 (1.53) | Delta |

"-" indicates no data for that field

following commercial extraction $\leq$ 30, compared to 1/4 replicates (25.0%) with Ct values > 30 (S1 Table in S1 File).

Following extraction with RNAES packets, all eluates with detectable SARS-CoV-2 RNA and sufficient remaining volume (n = 53 total) were tested in two separate Spike SNP rRT-PCRs. Out of those tested 52/53 (98.1%) had detectable signals in the Spike SNP assays (Table 1). Three genotypes were identified, of which the most common was K417variant/478K/501Y (31/53, 58.5%), consistent with Omicron variant. The two other genotypes detected in the eluates were K417 only (19/53, 35.8%), consistent with an ancestral lineage, and K417/452R/478K (2/53, 3.8%), consistent with Delta variant. Of the eluates for which Spike SNP testing was performed on the original samples (n = 38), all variant calls corresponded with those obtained after extraction with RNAES protocol.

## RNA stability at ambient temperature

A subset of five SARS-positive samples were chosen to evaluate the stability of dried RNA when stored on Fusion 5 membranes for up to one week at ambient temperature (Fig 2, S2 Table in S1 File). On day 1, 1/10 replicates had no detectable signal, indicating failed extraction. All other replicates had detectable positive Ct values across the remaining time points (Fig 2). The range in concentration of samples was 0.32 to 2.61 $\log_{10}$ copies/$\mu$L on day 0 and 0.09 to 2.41 $\log_{10}$ copies/$\mu$L on day 7 (S2 Table in S1 File). Overall, no significant difference in SARS-CoV-2 RNA concentration in the RNAES eluates (expressed in $\log_{10}$ copies/$\mu$L) was detected from day 0 (mean 1.7, SD 0.9) to day 1 (1.5, 0.7; p = 0.93), day 3 (1.5, 0.9; p = 0.95), and day 7 (1.1, 1.0; p = 0.36).

## Sequencing

SARS-CoV-2 full genome sequencing was performed for 13 representative samples spanning a range of Ct values from 23.5 to 33.6 (Table 2). Approximately 1 million total sequencing reads were generated from each sample (median 0.97 million, range 0.45–1.36 million), and complete SARS-CoV-2 genomes were assembled for most samples. Specifically, the 10 samples with SARS-CoV-2 Ct values < 30 yielded genomes with 99–100% coverage and a median depth of 1800-4900X. The 3 samples with SARS-CoV-2 Ct values > 30 yielded genomes with 85–97% coverage, which was sufficient for lineage classification. These results were very similar to results our group routinely obtains from samples with comparable Ct values extracted on two automated machines (KingFisher Apex and Abbott m2000sp, Table 2) [8].

## Discussion

The SARS-RNAES protocol successfully extracted SARS-CoV-2 RNA from residual swab samples at a cost of ~$0.08 per sample, and performance of the resulting eluates was similar to

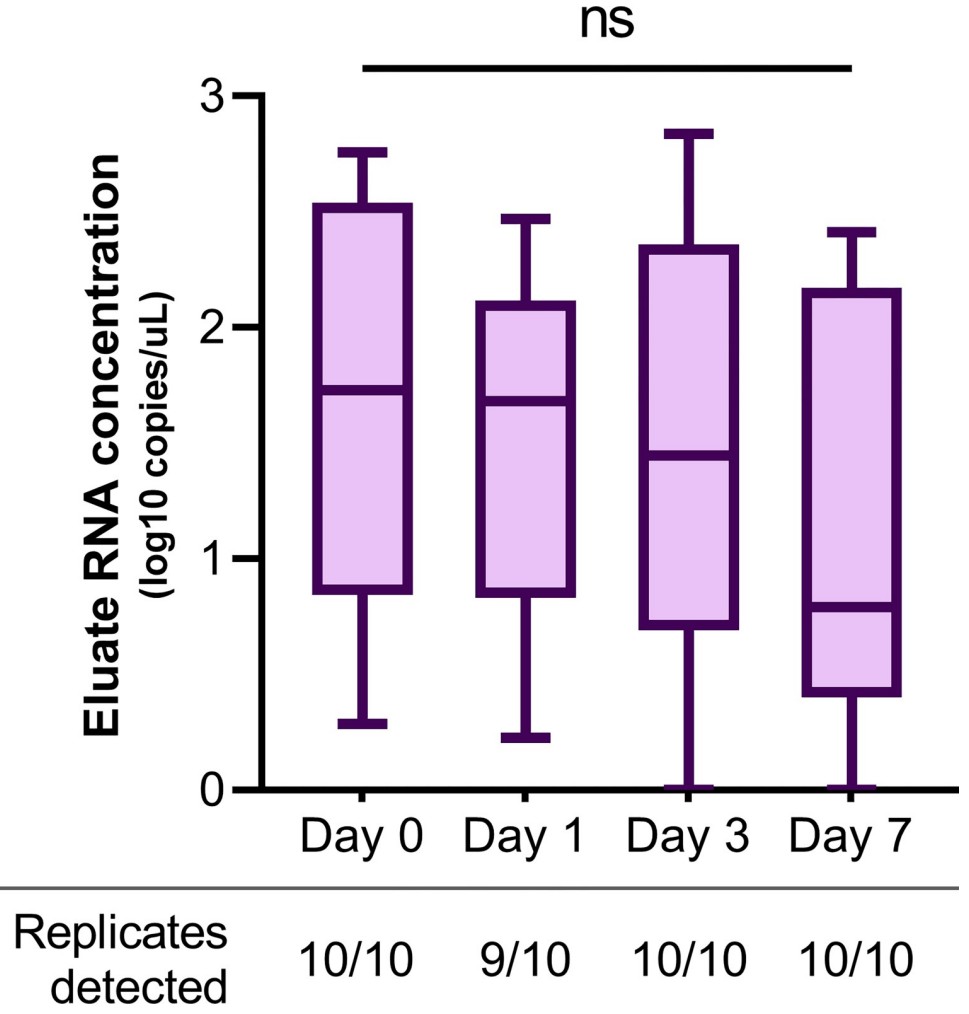

**Fig 2. Box and whisker plot of SARS-CoV-2 eluate RNA concentrations for samples extracted in RNAES packets with Fusion 5 membranes on day 0 or dried and stored on the membranes at ambient temperature for 1, 3, and 7 days after extraction.** No significant change in RNA concentration over time was found by ANOVA (displayed on graph) or by t-test comparisons of results on days 1, 3, and 7 versus day 0 ($p > 0.05$ for all comparisons). Whiskers extend from the maximum to minimum values.

those of commercial extraction robots, matched for Ct value, in Spike SNP genotyping and whole genome sequencing protocols.

Limited and inconsistent access to reagents for viral RNA extraction has resulted in the development of extraction-free methods for SARS-CoV-2 molecular detection. Such protocols utilize modified thermocycling conditions and additional master mix reagents to provide direct specimen testing, which is facilitated by relatively inhibitor-free primary clinical specimens [12–16]. While such techniques require less dedicated consumables and processing time, resulting nucleic acids cannot be applied to downstream molecular characterization of SARS-CoV-2 variants or further workup of negative cases. Moreover, extraction free methods require changes to laboratory biosafety practices and molecular workflow. In the current study, we demonstrate the suitability of the SARS-RNAES protocol for incorporation into SARS-CoV-2 genotyping or whole genome sequencing protocols. Performance of RNAES eluates was commensurate with those from expensive commercial robotic extraction systems, when matched

**Table 2. SARS-CoV-2 whole genome sequencing results following extraction in the RNAES protocol compared to two reference automated extraction protocols.**

| Sample ID | RNA extraction | Ct | Total # Reads | Median Genome Coverage | % Coverage <1x | % Coverage > = 10x | % Coverage > = 100x |
|---|---|---|---|---|---|---|---|
| **4546** | **RNAES** | **23.54** | **917,426** | **3,120** | **0** | **100** | **100** |
| EHC_C19_3973U | KingFisher Apex | 23.60 | 774,584 | 2,284 | 0 | 100 | 99 |
| EHC_C19_2672T | Abbott m2000sp | 23.40 | 921,096 | 2,440 | 0 | 100 | 100 |
| **4628b** | **RNAES** | **23.91** | **975,622** | **3,480** | **0** | **100** | **100** |
| EHC_C19_4423C | KingFisher Apex | 23.80 | 819,620 | 2,193 | 0 | 100 | 100 |
| EHC_C19_4028X | Abbott m2000sp | 23.80 | 897,420 | 2,566 | 0 | 100 | 99 |
| **1905** | **RNAES** | **24.42** | **788,444** | **2,556** | **0** | **100** | **100** |
| EHC_C19_4429I | KingFisher Apex | 24.40 | 800,206 | 2,519 | 0 | 100 | 100 |
| EHC_C19_2707C | Abbott m2000sp | 24.50 | 782,438 | 2,368 | 0 | 100 | 100 |
| **4628a** | **RNAES** | **24.60** | **778,698** | **2,780** | **0** | **100** | **99** |
| EHC_C19_4432L | KingFisher Apex | 24.70 | 944,770 | 2,369 | 0 | 100 | 100 |
| EHC_C19_4211Y | Abbott m2000sp | 24.60 | 938,200 | 2,962 | 0 | 100 | 99 |
| **7141** | **RNAES** | **25.97** | **1,476,272** | **4,921** | **0** | **100** | **100** |
| EHC_C19_4373E | KingFisher Apex | 25.90 | 1,163,252 | 4,378 | 0 | 100 | 100 |
| EHC_C19_2699U | Abbott m2000sp | 25.90 | 1,127,414 | 3,433 | 0 | 100 | 100 |
| **6076** | **RNAES** | **26.02** | **972,010** | **3,346** | **0** | **100** | **100** |
| EHC_C19_4438R | KingFisher Apex | 26.00 | 965,942 | 2,362 | 0 | 100 | 100 |
| EHC_C19_3776F | Abbott m2000sp | 26.10 | 1,059,202 | 3,737 | 0 | 100 | 100 |
| **274** | **RNAES** | **26.07** | **534,842** | **1,799** | **0** | **100** | **100** |
| EHC_C19_4444X | KingFisher Apex | 26.30 | 700,992 | 1,627 | 0 | 100 | 99 |
| EHC_C19_2628D | Abbott m2000sp | 26.00 | 549,078 | 1,607 | 0 | 100 | 100 |
| **1172** | **RNAES** | **27.53** | **1,168,060** | **3,753** | **0** | **100** | **99** |
| EHC_C19_4251M | KingFisher Apex | 27.40 | 1,108,946 | 1,958 | 0 | 99 | 97 |
| EHC_C19_2657E | Abbott m2000sp | 27.20 | 982,188 | 2,538 | 0 | 100 | 99 |
| **38** | **RNAES** | **29.68** | **1,209,944** | **3,100** | **0** | **100** | **99** |
| EHC_C19_3734P | KingFisher Apex | 29.80 | 976,210 | 2,348 | 0 | 99 | 98 |
| EHC_C19_2706B | Abbott m2000sp | 29.80 | 1,125,398 | 4,035 | 0 | 100 | 100 |
| **6478** | **RNAES** | **29.70** | **1,363,276** | **3,456** | **0** | **99** | **99** |
| EHC_C19_3960H | KingFisher Apex | 30.00 | 1,114,680 | 2,585 | 0 | 100 | 98 |
| EHC_C19_2706B | Abbott m2000sp | 29.80 | 1,125,398 | 4,035 | 0 | 100 | 100 |
| **39** | **RNAES** | **30.60** | **1,181,994** | **240** | **7** | **87** | **63** |
| EHC_C19_4379K | KingFisher Apex | 30.20 | 1,246,196 | 2,518 | 1 | 99 | 97 |
| EHC_C19_3890P | Abbott m2000sp | 30.50 | 1,081,672 | 1,868 | 0 | 100 | 99 |
| **43** | **RNAES** | **30.69** | **454,358** | **718** | **14** | **85** | **77** |
| EHC_C19_4100R | KingFisher Apex | 30.40 | 643,006 | 92 | 1 | 89 | 47 |
| EHC_C19_4083A | Abbott m2000sp | 30.50 | 611,038 | 1,392 | 0 | 99 | 96 |
| **9160** | **RNAES** | **33.60** | **1,396,922** | **2,834** | **3** | **95** | **92** |
| EHC_C19_3744Z | KingFisher Apex | 33.60 | 862,606 | 655 | 17 | 80 | 72 |
| EHC_C19_4355M | Abbott m2000sp | 33.40 | 1,154,262 | 593 | 1 | 96 | 81 |

on Ct values, in both the Spike SNP assay and a widely used whole genome sequencing protocol. Continuous identification of emerging variants has proven critical to understanding transmission patterns, viral evolution, and the clinical presentation of SARS-CoV-2 infections [17–19]. SARS-RNAES, therefore, provides an economical and safe solution for sourcing RNA extraction reagents while provide material for critical viral characterization.

Expansive development of novel molecular diagnostics has been integral for the timely detection of SARS-CoV-2 to initiate effective treatment and isolate those who may transmit

the virus. High demand for SARS-CoV-2 testing has made sourcing and maintaining the dedicated reagents and consumables for particular molecular platforms a major burden, thereby limiting their wide-scale implementation [5, 6]. The SARS-RNAES protocol is a simple, inexpensive method for the isolation of SARS-CoV-2 viral RNA, utilizing easy-to-source laboratory reagents and materials. Without the use of electric instrumentation, hazardous chemicals, and costly consumables, the SARS-RNAES protocol demonstrated successful detection of 93.3% of clinical samples tested. Of samples with initial Ct values ≤30, 98.2% were successfully extracted using this protocol. Notably, using the laboratory reference protocols described in this study, our group has demonstrated a significant association between SARS-CoV-2 nucleocapsid antigen and subgenomic RNA detection and Ct ≤30, indicating active viral replication and the potential for transmission [20, 21]. These data suggest that the SARS-RNAES protocol provides efficient RNA extraction from individuals at the highest risk to transmit in the community.

Finally, we examined the stability of extracted SARS-CoV-2 RNA on dried membranes for up 7 days at ambient temperature. Cold-chain requirements for sample collection, shipment, and storage has long since posed strict limitations on specimen handling. While the implications of poor storage conditions for RNA are well-established, a previous study highlights the importance of sample preparation and storage conditions for successful detection of SARS-CoV-2, reporting that improper conditions can lead to misclassification of up to 10.2% of SARS-CoV-2-positive cases [22]. Here we provide a successful and sustainable technique that addresses the limitations of current specimen handling requirements while maintaining accurate detection.

Limitations to the current study include the efficient extraction of SARS-CoV-2 RNA only from RNAES packets prepared with Fusion 5 membranes, in contradistinction to adequate extraction of blood-borne viruses on multiple membrane types [7]. This additionally impacts the window of nucleic acid stability at ambient temperatures, as we have found that RNA on Fusion 5 membranes remains stable for shorter periods of time compared to glass-fiber membranes. Further studies should examine these interactions more closely.

This safe and cost-effective technique was established to address key limitations in current protocols for nucleic acid extraction and storage. SARS-RNAES balances the competing demands placed on laboratories to maintain biosafe laboratory practices, ensure a consistent supply chain of reagents, and provide high-quality RNA for a variety of molecular applications. With reproducible results across a range of virus concentrations, the SARS-RNAES protocol could help increase SARS-CoV-2 diagnostic testing and monitoring for emerging variants in resource-constrained communities.

## Supporting information

**S1 File. S1 and S2 Tables displaying Ct values following RNA extraction with the SARS-RNAES protocol and a commercial extraction robot and the average concentration of SARS-CoV-2 RNA (log$_{10}$ copies/μL) in duplicate RNAES extractions following ambient temperature storage for 0-, 1-, 3-, and 7-days post extraction.**
(PDF)

## Acknowledgments

We thank all members of the research team and the participants and their family members who have contributed to ongoing studies. We also thank Ali Haider, Maxwell Su, Jaewon Shin, and Victoria Stittleburg at Emory University for their assistance over the course of this project.

## Author Contributions

**Conceptualization:** Sarah Hernandez, Jesse J. Waggoner.

**Data curation:** Sarah Hernandez, Taz Azmain.

**Formal analysis:** Sarah Hernandez, Phuong-Vi Nguyen, Taz Azmain, Anne Piantadosi, Jesse J. Waggoner.

**Funding acquisition:** Jesse J. Waggoner.

**Investigation:** Sarah Hernandez, Phuong-Vi Nguyen.

**Methodology:** Sarah Hernandez, Jesse J. Waggoner.

**Project administration:** Jesse J. Waggoner.

**Supervision:** Jesse J. Waggoner.

**Writing – original draft:** Sarah Hernandez, Jesse J. Waggoner.

**Writing – review & editing:** Sarah Hernandez, Taz Azmain, Anne Piantadosi, Jesse J. Waggoner.

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
