## [Decision Letter · Decision Letter 0]

16 Nov 2022

PONE-D-22-24969SARS-CoV-2 genotyping and sequencing following a simple and economical RNA Extraction and Storage protocolPLOS ONE

Dear Dr. Waggoner,

Thank you for submitting your manuscript to PLOS ONE. After careful consideration, we feel that it has merit but does not fully meet PLOS ONE’s publication criteria as it currently stands. Therefore, we invite you to submit a revised version of the manuscript that addresses the points raised during the review process.

We look forward to receiving your revised manuscript.

Kind regards,

Ahmed S. Abdel-Moneim, Ph.D.

Academic Editor

PLOS ONE

Journal Requirements:

“This research was supported by an award from the Doris Duke Charitable Foundation (Clinical Scientist Development Award 2019089, JJW).”

“This research was supported by an award from the Doris Duke Charitable Foundation (Clinical Scientist Development Award 2019089, JJW; https://www.ddcf.org/). The funders had no role in study design, data collection and analysis, decision to publish, or preparation of the manuscript.  “

5. We note you have included a table to which you do not refer in the text of your manuscript. Please ensure that you refer to Table 1 in your text; if accepted, production will need this reference to link the reader to the Table.

Reviewers' comments:

Reviewer's Responses to Questions

**Comments to the Author**

1. Is the manuscript technically sound, and do the data support the conclusions?

Reviewer #1: Yes

Reviewer #2: Yes

2. Has the statistical analysis been performed appropriately and rigorously? 

Reviewer #1: Yes

Reviewer #2: Yes

3. Have the authors made all data underlying the findings in their manuscript fully available?

Reviewer #1: Yes

Reviewer #2: Yes

4. Is the manuscript presented in an intelligible fashion and written in standard English?

Reviewer #1: Yes

Reviewer #2: Yes

5. Review Comments to the Author

Reviewer #1: The authors have done an interesting work and have tried to come up with an alternative and economical solution for RNA extraction and short-term storage at ambient temperature for SARS-CoV-2. Additionally, they have also tried to evaluate the quality of the RNA isolated by their method for SNP detection and WGS.

The paper looks well written and structured, but I feel that it needs to shift the focus of introduction and discussion from the RNA extraction to SARS-CoV-2 genotyping and sequencing. The article in its current form gives a feeling of redundancy and replication of their recently published article “Simple and Economical Extraction of Viral RNA and Storage at Ambient Temperature” and little importance to the novel part of the work. Also, it would be better if the extraction protocol is explained properly in the method section of the article because RNA extraction method is one of the aims of work. Other than the writing part, I have few other concerns and questions related to the experiments.

I am surprised that why GF/D membranes, which had best yield for DENV-positive serum samples, didn’t work at all for the SARS-CoV-2 nasopharyngeal samples. The authors have tried to find the reason, but I have few general questions related to it. Was there no RNA in the elution or the PCR didn’t work? How many times did they repeat the experiment with all the three membranes? Did they use the same set of samples and conducted the experiments at the same time?

One of the major claims of their work is the stability of the RNA at ambient temperature for 7 days. It would be worth knowing the difference in the Ct values for the samples eluted at different time points and their comparison with the MagMaxViral RNA Isolation Kit on day 0, for its practical applicability.

Have they tested the maximum time limit for successful RNA extraction?

The authors have shown that Ct values < 30 yielded genomes with 99-100% coverage. Was there any correlation between the RNA concentration and the Ct values of the samples?

Reviewer #2: First, I would like to congratulate the authors for the manuscript, it is well written and deals with a scientific relevant subject. This protocol may help laboratories around the world with limited resources to rapidly detect SARS-CoV-2 in nasopharyngeal samples, specially when facing emergencial situations.

The main objective of the research was clear to me and the performed tests showed satisfying results. The comparison with commercial kits revealed the protocol to be consistent and the limitations of the work were properly highlighted. The only issue I had was with Table 1 "Genotypes detected in the Spike SNP assay following RNAES extraction.", it seems it is not well formatted. I kindly ask for the authors to take a look at it.

Best regards.

---

## [Author Response · Author response to Decision Letter 0]

8 Dec 2022

Our response to reviewers document has been uploaded with the revision. We have addressed each comment in turn. In addition, we have made all changes requested to conform to PLOS ONE requirements. The funding statement, as originally generate in the online system, is accurate.

---

## [Decision Letter · Decision Letter 1]

4 Jan 2023

SARS-CoV-2 genotyping and sequencing following a simple and economical RNA Extraction and Storage protocol

PONE-D-22-24969R1

Dear Dr. Waggoner,

We’re pleased to inform you that your manuscript has been judged scientifically suitable for publication and will be formally accepted for publication once it meets all outstanding technical requirements.

Kind regards,

Ahmed S. Abdel-Moneim, Ph.D.

Academic Editor

PLOS ONE

---

## [Editor Report · Acceptance letter]

6 Jan 2023

PONE-D-22-24969R1 

SARS-CoV-2 genotyping and sequencing following a simple and economical RNA Extraction and Storage protocol 

Dear Dr. Waggoner:

I'm pleased to inform you that your manuscript has been deemed suitable for publication in PLOS ONE. Congratulations! Your manuscript is now with our production department. 

Kind regards, 

on behalf of

Prof. Ahmed S. Abdel-Moneim 

Academic Editor

PLOS ONE